# Inhibitory Effect of *Salvia miltiorrhiza* Extract and Its Active Components on Cervical Intraepithelial Neoplastic Cells

**DOI:** 10.3390/molecules27051582

**Published:** 2022-02-27

**Authors:** Xuejiao Leng, Hongfei Kan, Qinhang Wu, Cunyu Li, Yunfeng Zheng, Guoping Peng

**Affiliations:** College of Pharmacy, Nanjing University of Chinese Medicine, 138 Xianlin Avenue, Qixia District, Nanjing 210023, China; leng.xuejiao2008@163.com (X.L.); 20200829@njucm.edu.cn (H.K.); wuqinhang@njucm.edu.cn (Q.W.); licunyuok@163.com (C.L.); zyunfeng88@126.com (Y.Z.)

**Keywords:** cervical intraepithelial neoplasia, salvianolic acid A, oxysophoridine, synergistic effect, G2/M phase arrest

## Abstract

The effective treatment of cervical intraepithelial neoplasia (CIN) can prevent cervical cancer. *Salvia miltiorrhiza* is a medicinal and health-promoting plant. To identify a potential treatment for CIN, the effect of *S. miltiorrhiza* extract and its active components on immortalized cervical epithelial cells was studied in vitro. The H8 cell was used as a CIN model. We found that *S. miltiorrhiza* extract effectively inhibited H8 cells through the CCK8 method. An HPLC–MS analysis revealed that *S. miltiorrhiza* extract contained salvianolic acid H, salvianolic acid A, salvianolic acid B, monomethyl lithospermate, 9‴-methyl lithospermate B, and 9‴-methyl lithospermate B/isomer. Salvianolic acid A had the best inhibitory effect on H8 cells with an IC_50_ value of 5.74 ± 0.63 μM. We also found that the combination of salvianolic acid A and oxysophoridine had a synergistic inhibitory effect on H8 cells at molar ratios of 4:1, 2:1, 1:1, 1:2, and 1:4, with salvianolic acid A/oxysophoridine = 1:2 having the best synergistic effect. Using Hoechst33342, flow cytometry, and Western blotting analysis, we found that the combination of salvianolic acid A and oxysophoridine can induce programmed apoptosis of H8 cells and block the cell cycle in the G2/M phase, which was correlated with decreased cyclinB1 and CDK1 protein levels. In conclusion, S. miltiorrhiza extract can inhibit the growth of H8 cells, and the combination of salvianolic acid A (its active component) and oxysophoridine has a synergistic inhibitory effect on H8 cells and may be a potential treatment for cervical intraepithelial neoplasia.

## 1. Introduction

Cervical cancer is the fourth most common malignant tumor in women worldwide, with 530,000 new cases and 270,000 deaths every year [1]. At present, the main methods of managing cervical cancer are prevention via the HPV vaccine before disease diagnosis and surgical treatment after diagnosis [2]. However, the application of the HPV vaccine in non-developed countries has not been popularized [3,4]. Surgical resections after the diagnosis of the disease also cause permanent damage to the human body [5,6]. The progression of the HPV infection to cervical cancer takes decades. During this process, cervical intraepithelial neoplasia (CIN) always occurs, which is equivalent to the clinical “precancerous” stage [2,7]. At present, CIN is mostly treated through surgery. The recommended treatments are regular examinations and follow-ups for patients with CIN1 and therapeutic excision by either a cold knife conization or a loop electrosurgical excision procedure for patients with CIN2 and CIN3 [8]. However, surgical resection for CIN leads to long-term reproductive diseases, and untimely treatment may damage the reproductive system and has the risk of transforming into cervical cancer [8,9]. Therefore, drug treatments of CIN should be studied. The use of a drug treatment can reduce the pain caused by surgery and prevent the occurrence of cervical cancer.

*Salvia miltiorrhiza*, also known as danshen, is a well-known Chinese herbal medicine used in human medicine for its cardiovascular and cerebrovascular protective effects [10] and as a health-promoting food that improves sleep and calms nerves [11]. *S. miltiorrhiza* has been used to treat cardiovascular diseases in China and other Asian countries for hundreds of years [12]. The chemical constituents of *S. miltiorrhiza* are well identified, including more than 30 lipophilic compounds with diterpene quinone structures and more than 50 hydrophilic compounds with phenolic acid structures [13]. Salvianolic acids are some of the most important bioactive components of *S. miltiorrhiza* and have antioxidant, anti-inflammatory, antibacterial, antitumor, and other biological qualities [11]. Salvianolic acids have also been approved by the Chinese FDA for the treatment of ischemic strokes [14].

Salvianolic acids have been shown to possess antitumor activities [15]. Among them, salvianolic acid A can inhibit the growth of lung carcinoma [16], oral squamous cell carcinoma [17], cervical cancer [18], and breast cancer [19]. Salvianolic acid B can inhibit the growth of head and neck carcinoma [20], colorectal cancer [21], cervical cancer [22], glioma, and ovarian cancer [23] by inducing apoptosis, halting the cell cycle, and adjusting metastasis by targeting multiple deregulated signaling networks of cancer [15]. Salvianolic acid A (SAA) and salvianolic acid B are effective components of *S. miltiorrhiza* and both can inhibit the growth of cervical cancer cells [18,22]. Therefore, we hypothesized that the active components of *S. miltiorrhiza* could also effectively treat CIN and studied the inhibitory effect of the *S. miltiorrhiza* extract on H8 cells. We found that the *S. miltiorrhiza* extract inhibited the activity of H8 cells. The combination of SAA (its active components) and oxysophoridine (OSR) exerted a synergistic inhibitory effect on H8 cells. This finding suggests that a combination of SAA and OSR may be a potential treatment for CIN.

## 2. Methods

### 2.1. Materials

*S. miltiorrhiza* medicinal slices were purchased from the Anhui Renpu Pharmaceutical Co. (Anhui, China). The AB-8 macroporous adsorption resin (lot no. 312L011) was obtained from Beijing Solebo Technology Co. (Beijing, China). SAA, salvianolic acid B, and OSR (purity ≥ 98.0%) were purchased from Shanghai Yuanye Biotechnology (Shanghai, China). CCK8 was purchased from Beyotime (Shanghai, China). Hoechst 33342, propidium iodide (PI), and RNase were purchased from Keygen Biotech (Shanghai, China). Anti-CyclinB1 and anti-CDK1 antibodies were purchased from Boster Biotechnology (Shanghai, China). Analytical grade hydrochloric acid and other chemicals were purchased from Sinopharm Group Chemical Reagent Co., Ltd. (Shanghai, China).

### 2.2. Preparation of Extracts

*S. miltiorrhiza* extract (SME): We sifted 50 g of *S. miltiorrhiza* powder through a no. 2 sieve, extracted it with 1000 mL of water, and left it overnight. After adjusting the pH of the filtrate to 3–4 with 1% HCl, it was passed through an AB-8 macroporous resin (100 mL) and eluted with 500 mL of water, 500 mL of 6% ethanol, and 500 mL of 35% ethanol. The final eluent was collected and then the extract was recovered at 40 °C, concentrated in a vacuum, and freeze dried to obtain a fine loose powder.

### 2.3. Detection Conditions of HPLC-MS

The HPLC chromatogram of the SME was obtained using a Waters Series 2695 liquid chromatography system (Waters Technologies, Milford, MA, USA). In accordance with the method of Zhai et al. [24], the column used was an ODS-2 C_18_ column (250 × 4.6 mm, 5 μm), and the samples were separated using a gradient mobile phase consisting of acetonitrile (A) and 0.2% (*v*/*v*) formic acid water (B). The gradient conditions were as follows: 10–20% A at 0–10 min and 20–45% A at 10–30 min. The flow rate was set at 1.0 mL/min, the detection wavelength was 281 nm, the column temperature was 25 °C, the sample concentration was 1 mg/mL, and the injection volume was 10 μL.

For mass spectrometry, an AB SCIEX Triple TOFTM5600 high-resolution mass spectrometer (SCIEX, Foster City, CA, USA) was used for detection under the ESI negative ion mode. The parameters were as follows: *m*/*z* scanning range of 50–1500 Da; spray voltage of 5500 V, ion spray no-load voltage of −4500 V, de-clustering voltage of −100 V, collision energy of −10 V, ion-source temperature of 550 °C, atomizer pressure of 60 psi, auxiliary heater pressure of 60 psi, and air curtain pressure of 40 psi.

### 2.4. Cell Lines and Culture

The H8 cell line was provided by the Institute of Basic Medicine, Union Medical University, Chinese Academy of Medical Sciences. It was an immortalized HPV16-positive human cervical epithelial cell model. Owing to its unique characteristics, it had great significance in the study of carcinogenesis induced by HPV-16. However, it failed to cause carcinogenesis within six months after injection into nude mice [25,26]. H8 cells were cultured in Roswell Park Memorial Institute medium 1640 (HyClone, Logan, UT, USA) supplemented with 10% (*v*/*v*) fetal bovine serum (Gibco, MA, USA) and 1% antibiotics (100 U/mL penicillin and 100 μg/mL streptomycin) (Gibco, MA, USA) in a 37 °C humidified atmosphere containing 95% air and 5% CO_2_.

### 2.5. Cell Viability Assay

The cell viability assay was performed using the CCK8 method. The cells were seeded in 96-well plates (1 × 10^5^ cells per well) and cultured. Drugs at a series of concentrations were added to each well. After 24 h of incubation, the cell viability was determined by the CCK8 assay. The inhibition ratio was calculated using the following equation:$$\mathrm{Cell}\;\mathrm{inhibitory}\;\mathrm{ratio}=1-\frac{\mathrm{As}-\mathrm{Ab}}{\mathrm{Ac}-\mathrm{Ab}}*100\%$$
where As is the absorbance of the experimental well (including cell, medium, CCK8, and drug), Ac is the absorbance of the control well (containing cells, medium, and CCK8), and Ab is the absorbance of the blank well (containing medium and CCK8).

### 2.6. Synergistic Cytotoxicity Effect of Drug Combination

The synergy between the two drugs was evaluated using the Chou–Talalay method to calculate the combination index. The CI was calculated using the following equation [27,28]:$$\mathrm{CI}(x)=\frac{{\left(D\right)}_{1}}{{\left(\mathrm{Dx}\right)}_{1}}+\frac{{\left(D\right)}_{2}}{{\left(\mathrm{Dx}\right)}_{2}}$$
where (D)_1_ and (D)_2_ are the concentrations of a single drug after combination treatment that inhibits x% of cell growth, and (Dx)_1_ and (Dx)_2_ are the concentrations of a single drug alone that inhibits x% of cell growth. Combination index (CI) values greater than 1 or less than 1 indicate antagonism or synergism of drug combinations, respectively.

The dose-reduction index (DRI) or reversal ratio (or cytotoxicity-enhancement ratio) is a measure of the number of times the dose may be reduced compared with the doses of each drug alone. The DRI was calculated as (DRI)_1_ = (Dx)_1_/(D)_1_ and (DRI)_2_ = (Dx)_2_/(D)_2_. DRI > 1 is beneficial, indicating synergism [29,30].

### 2.7. Fluorescence Morphology Assay

Cells were seeded onto six-well culture plates and treated with 0, 0.5, 1, and 2 μmol/L AO (SAA/OSR = 1:2) for 24 h. The cells were washed with PBS, stained with Hoechst 33342 (1 μg/mL in PBS) for 30 min at 4 °C, and examined under a fluorescence microscope (Leica DMIRB).

### 2.8. Cell Cycle Detection by Flow Cytometry

H8 cells were inoculated in six-well plates at a density of 4 × 10^5^ cells/well. The cells were treated with 0, 0.5, 1, and 2 μmol/L AO (SAA/OSR = 1:2) for 24 h after cell adhesion. After trypsin digestion, the cells were centrifuged at 1000 r/min for 5 min. The cells were washed with 4 °C precooled PBS buffer, mixed with precooled 75% ethanol, fixed overnight at 4 °C, and washed again with precooled PBS buffer. After centrifugation, the PI staining solution (0.5 mL) containing RNase A was added to the suspensions of H8 cells. All samples were incubated at 37 °C for 30 min and analyzed using an 18-color flow cytometer (LSR II, BD Biosciences, CA, USA). The data were analyzed using FlowJo 8.6 software (TreeStar, Woodburn, OR, USA).

### 2.9. Western Blot Analysis

The cells (5 × 10^5^ cells/well) were seeded in six-well plates. After 12 h of incubation, the cells were treated with 0.5, 1, and 2 μmol/L AO for 24 h. After treatment with drugs, whole-cell proteins were obtained using a protein extraction kit, and concentrations were determined by a BCA assay (Pierce, Rockford, IL, USA). For protein detection, cell lysates containing 30 μg of protein were separated by size on a 10% SDS–Tricine gel (Invitrogen, Carlsbad, CA, USA) and transferred onto polyvinylidene difluoride membranes (Millipore, Bedford, MA, USA). Nonspecific binding sites were blocked by incubation in blocking buffer (1 × TBST with 5% (*w*/*v*) free-fat dry milk) for half an hour at room temperature, followed by incubation with primary antibodies overnight at 4 °C. After the primary antibody incubation, the membranes were washed three times with TBST (10 min each at room temperature) and incubated with the corresponding HRP-conjugated secondary antibody for 2 h at room temperature. The blots were washed three times with TBST, and bands were visualized on a camera using the West Femto and Pico ECL detection reagent (Life Technologies, Carlsbad, CA, USA). β-Actin was used as a loading control. ImageJ software (National Institutes of Health, Bethesda, MD, USA) was used to measure the densities, which were normalized to that of β-actin.

### 2.10. Statistical Analyses

All experimental data are expressed as means ± standard deviations. Statistical analyses were performed using SPSS Statistics 17.0 software to evaluate the differences between groups, which were considered significant at * *p* < 0.05; ** *p* < 0.01; and *** *p* < 0.001. All data represent the means of triplicate measurements.

## 3. Results

### 3.1. Inhibitory Effect of the SME on H8 Cells

To investigate the effect of the SME on H8 cells, H8 cells were treated with SME at the indicated concentrations for 24 h. As shown in Figure 1, the IC_50_ value of SME on H8 cells was 84.31 ± 0.54 μg/mL.

### 3.2. HPLC–MS Identification of the SME

To analyze the composition of the SME, the identification of components in the spectrum was compared and confirmed by mass spectrometry fragmentation of high-purity single components and UV full-wavelength absorption and was verified in combination with the literature data. As shown in Figure 2, the HPLC chromatogram of SME had six peaks. After analysis and consulting the relevant literature, the six components were confirmed as salvianolic acid H, SAA, salvianolic acid B, monomethyl lithospermate, 9‴-methyl lithospermate B, and 9‴-methyl lithospermate B/isomer, as shown in Table 1.

### 3.3. Inhibitory Effect of the Active Components of the SME on H8 Cells

In the SME, the contents of SAA and salvianolic acid B can inhibit the cell growth of H8, and their IC_50_ values were 5.74 ± 0.63 and 41.76 ± 3.87 μM, respectively. The IC_50_ value of SAA on H8 cells was significantly lower than that of salvianolic acid B; that is, the inhibitory effect of SAA on H8 cells was better than that of salvianolic acid B. We also found that OSR effectively inhibited the proliferation of H8 cells and that SAA and OSR exerted synergistic effects on H8 cells. The chemical structures of SAA and OSR are shown in Figure 3A. SAA exerted an inhibitory effect on H8 cells at 1 μM, and the maximum inhibition rate reached 94.6% ± 1.4%. The inhibition rate of OSR on H8 cells was 63.24% ± 2.57% at 32 μM; however, the inhibition rates did not subsequently increase between 32 and 128 μM, as shown in Figure 3B.

The synergistic effects of SAA and OSR on H8 cells were studied by the Chou–Talalay method [27,28,29,30]. Figure 3C shows that the inhibitory effect of SAA combined with OSR on H8 cells was stronger than that of SAA alone. The synergistic indices CI and DRI are shown in Figure 3D and Table 2. When Fa = 0.5 (inhibition rate = 50%) and SAA/OSR = 1:2, we obtained the lowest CI of 0.59, which had a relatively good synergistic effect. The IC_50_ value of SAA decreased from 5.74 ± 0.63 μM to 1.87 ± 0.37 μM, and the IC_50_ value of OSR decreased from 13.70 ± 1.71 μM to 3.73 ± 0.74 μM (Table 3). The DRI values were greater than 1, indicating synergism.

We used a combination of SAA and OSR for further studies. SAA/OSR = 1:2, abbreviated as AO; the concentration of SAA was used to express the concentration of AO. For example, 1 μM AO was 1 μM SAA plus 2 μM OSR.

### 3.4. Effect of AO on H8 Cell Apoptosis and the Cell Cycle

Hoechst 33342 can penetrate the cell membrane and enter normal cells, and apoptotic cells then combine with DNA. After staining, the living cells were dark blue and the normal apoptotic cells were bright blue [34,35]. As shown in Figure 4A, cells in the blank group were dark blue, whereas those in the treatment group were dark blue and bright blue. With increasing AO concentration, the number of bright blue cells and the number of normal apoptotic cells increased. Therefore, AO could induce programmed apoptosis in H8 cells. Cytometric analysis revealed that with an increased AO concentration, the G1 peak decreased and the G2 peak increased in the cell cycle of H8 cells, as shown in Figure 4B,C. These results suggest that AO inhibits cell proliferation via G2/M phase arrest.

### 3.5. Effect of AO on the Expression of Cell-Cycle-Related Proteins in H8 Cells

To understand the molecular mechanism underlying the effect of AO on cell cycle progression, we investigated the effect of AO on the expression of key proteins (including CDK1 and cyclin B1) in the G2/M transition. The Western blot results of the AO-treated (0.5, 1, and 2 μm AO) and untreated cells are shown in Figure 5A,B. Decreased levels of cyclin B1 and CDK1 protein were observed 24 h after treatment, suggesting that the downregulation of the expression of G2-phase-regulating proteins may have contributed to the AO-mediated cell cycle arrest of H8 cells.

## 4. Discussion

Despite the implementation of HPV vaccination strategies and advances in chemoradiation and immunotherapy, cervical cancer remains a major health concern [36]. The treatment of cervical intraepithelial neoplasia with drugs can reduce the pain caused by surgery and prevent the occurrence of cervical cancer.

S. *miltiorrhiza* is commonly used in traditional Chinese medicine as a health-promoting food [2]. SAA and salvianolic acid B, the active components of *S. miltiorrhiza*, can inhibit the growth of cervical cancer cells [18,22]. Recent studies have shown that SAA could protect normal cells from damage, inhibit cardiomyocyte apoptosis in response to I/R [37], and ameliorate the human ovarian granulosa cell damage and human renal tubular epithelial cell injury induced by H_2_O_2_ [38,39]. We studied the extract of *S. miltiorrhiza* to identify its effective components for treating cervical intraepithelial neoplasia. The inhibitory effects of SME and its active components on H8 cells were determined using the CCK8 method. The results showed that SME effectively inhibited the proliferation of H8 cells. Through an HPLC–MS analysis, we found that the SME contained salvianolic acid H, SAA, salvianolic acid B, monomethyl lithospermate, 9‴-methyl lithospermate B, and 9‴-methyl lithospermate B/isomer. SAA and salvianolic acid B inhibited the growth of H8 cells, with IC_50_ values of 5.74 ± 0.63 and 41.76 ± 3.87 μM, respectively. The IC_50_ value of SAA in H8 cells was significantly lower than that of salvianolic acid B. *S. miltiorrhiza* is a well-known Chinese traditional medicine with complex components. The main salvianolic acid component of *S. miltiorrhiza* is salvianolic acid B [40,41,42], and many researchers have studied salvianolic acid B [22,41,42]. However, the main component is not necessarily the most effective component. In the present study, SAA had a stronger inhibitory effect on H8 cells than salvianolic acid B. Therefore, we selected SAA for further research. Due to the low content of SAA in *S. miltiorrhiza**,* some researchers transformed salvianolic acid B to SAA by using different reaction conditions, including pH, temperature, pressure, humidity [43], or subcritical water extraction [44] to increase the yield of SAA. Oxysophoridine (OSR), an alkaloid extracted from *Sophora alopecuroides* L, has antioxidant, anti-inflammatory, and antiapoptotic effects on spinal cord injury [45]. It also has protective effects against ischemic brain damage [46]. Considering that most traditional Chinese medicine prescriptions are effective, we selected OSR as the research object to identify drugs that can cooperate with SAA. Our findings demonstrated that OSR could effectively inhibit the proliferation of H8 cells, and SAA and OSR exerted a strong synergistic effect on H8 cells; the best synergistic effect was observed when SAA/OSR = 1:2. Fluorescence morphology and flow cytometry analyses showed that the combination of SAA and OSR caused programmed apoptosis and G2/M phase arrest in H8 cells in a dose-dependent manner. Cell cycle disorders are known markers of tumor cells. Cell cycle proteins may be promising targets for cancer therapy [47]. CyclinB1 is one of the main regulatory elements at the G2/M restriction point in the cell cycle. CDK1, also known as cdc2 (cell division cycle gene), encodes the CDK1 protein. CyclinB-CDK1 is an important component of the M-phase promoter, which regulates the G2/M phase transition [48,49]. Herein, we found that AO can significantly reduce the content of cyclinB1 and CDK1, inhibit the G2/M phase transition of H8 cells, block the cells in the G2/M phase, and prevent H8 cells from entering the mitotic phase. Therefore, AO blocks cells in the G2/M phase by downregulating cyclin B1 and CDK1 protein levels. Thus, the combination of SAA and OSR may be a potential treatment for CIN. However, this study has some limitations. Considering that few studies have reported the use of drugs for the treatment of CIN, we did not find a suitable reference drug for use in comparison with the test drug in this study. In addition, at a later stage, we need to conduct a comparative research study on single and combined drug use. We also need to further study the mechanism of combined drug use. Safety studies and clinical research need to be conducted to determine whether the combination of SAA and OSR can be used safely. Therefore, drug research for the treatment of CIN still has a long way to go.

## 5. Conclusions

SME can inhibit the activity of H8 cells. SAA has a better inhibitory effect on H8 cells than the other active components of the SME. The combination of SAA and OSR (SAA/OSR = 1:2) exerted a synergistic inhibitory effect on H8 cells, and the cell cycle arrest induced by this combination was associated with decreased CDK1 and cyclinB1 protein levels. Drug treatment of CIN can reduce the injury and fear caused by surgery and prevent the occurrence of cervical cancer. The combination of SAA and OSR can inhibit CIN cells, which lays a solid foundation for the further study of the treatment of CIN.

## Figures and Tables

**Figure 1 molecules-27-01582-f001:**
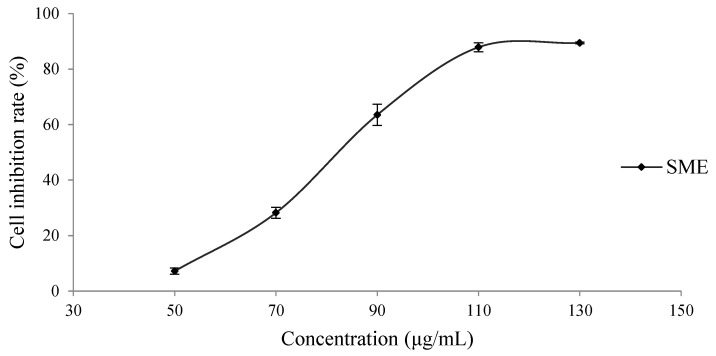
Inhibitory effect of the SME on H8 cells.

**Figure 2 molecules-27-01582-f002:**
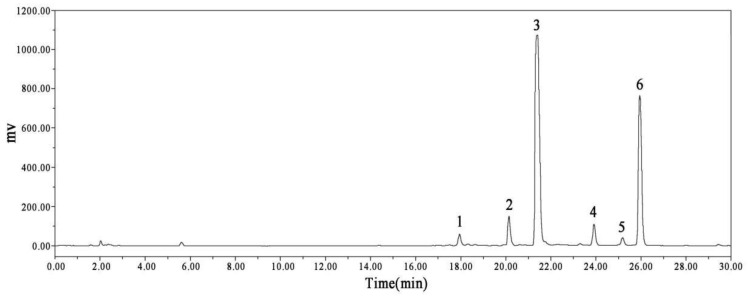
HPLC chromatogram of the SME.

**Figure 3 molecules-27-01582-f003:**
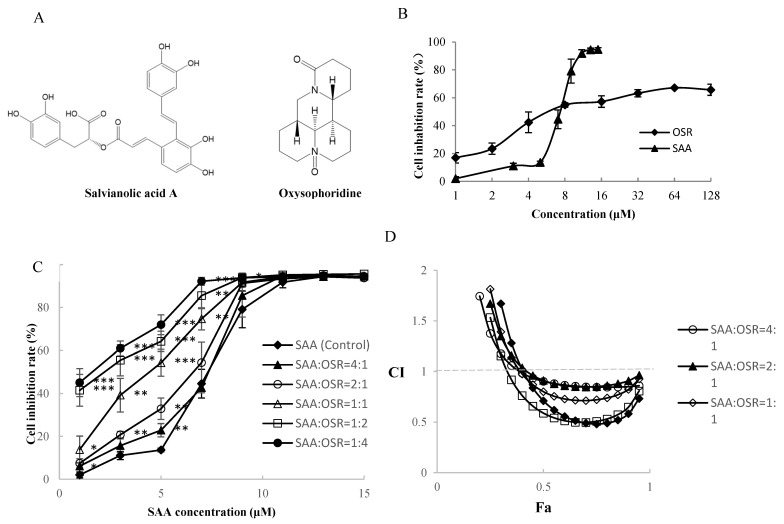
Analysis of synergism of salvianolic acid A (SAA) and oxysophoridine (OSR) on H8 cells. (**A**) Chemical structure of salvianolic acid A (SAA) and oxysophoridine (OSR). (**B**) Inhibitory effect of SAA and OSR on H8 cells. (**C**) Inhibitory effect of different proportions of SAA and OSR on H8 cells. (**D**) Plots of Fa-CI in different ratios by the Chou–Talalay method; (* *p* < 0.05, ** *p* < 0.01, *** *p* < 0.001).

**Figure 4 molecules-27-01582-f004:**
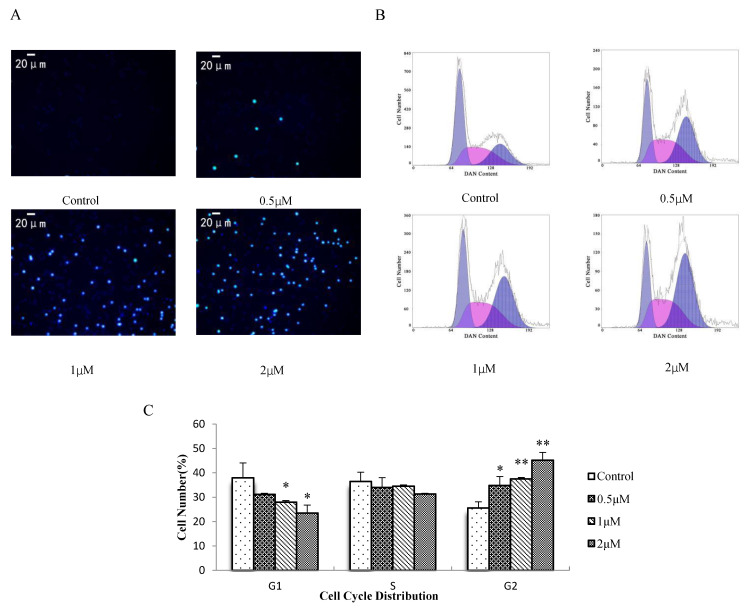
Analysis of cell fluorescence morphology and the cell cycle. (**A**) Fluorescence staining of nuclei in AO-treated and untreated cells using Hoechst 33342. (**B**) H8 cells were treated with different concentrations of AO and analyzed at 24 h by flow cytometry. (**C**) Percentage of cells in the indicated phases of cell cycle; (* *p* < 0.05, ** *p* < 0.01). SAA/OSR = 1:2 abbreviated as AO, which expressed the concentration of SAA as the concentration of AO, for example, 1 μM AO is 1 μM SAA plus 2 μM OSR.

**Figure 5 molecules-27-01582-f005:**
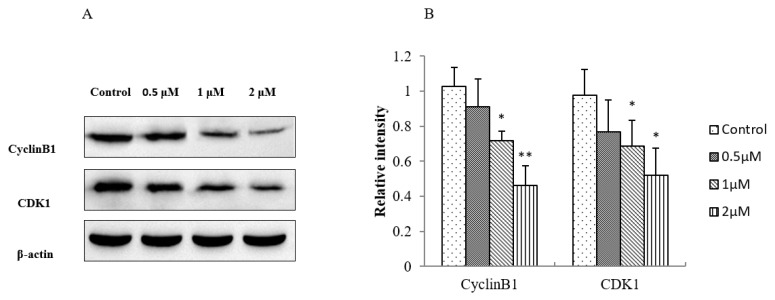
Western blot analysis of protein extracts obtained from H8 cells treated with 0, 0.5, 1, and 2 μmol/L AO. (**A**) Western blot analysis of CyclinB1 and CDK1 expression in H8 cells treated with 0, 0.5, 1, and 2 μmol/L AO. (**B**) Relative intensity values of CyclinB1 and CDK1 normalized by β-actin; (* *p* < 0.05, ** *p* < 0.01). SAA/OSR = 1:2 abbreviated as AO, which expressed the concentration of SAA as the concentration of AO, for example, 1 μM AO is 1 μM SAA plus 2 μM OSR.

**Table 1 molecules-27-01582-t001:** HPLC–MS identified in the typical base peak chromatogram of the SME samples.

Peak No.	Retention (min)	Molecular Formula	Measured Mass *m*/*z* (Error, ppm)	MS/MS Fragments Ions	Relative Quantification (%) ^(a)^	Identification
1	18.01	C_27_H_22_O_12_	538.1018 (−2.5)	538, 340, 295, and 185	1.96	salvianolic acid H [31]
2	20.20	C_26_H_22_O_10_	493.1118 (0.3)	493, 295, and 185	4.72	salvianolic acid A [31]
3	21.68	C_36_H_30_O_16_	717.1461 (−5.0)	717, 519, 339, 321, and 295	57.62	salvianolic acid B [32]
4	23.76	C_28_H_24_O_12_	551.1171 (−4.9)	551, 339, and 321	3.60	monomethyl lithospermate [32]
5	25.15	C_37_H_32_O_16_	731.1618 (−4.9)	731, 533, 339, 321, and 295	1.52	9‴-methyl lithospermate B [33]
6	25.77	C_37_H_32_O_16_	731.1618 (−5.1)	731, 533, 507, 339, and 295	30.58	9‴-methyl lithospermate B/isomer [33]

^(a)^ Relative quantification (%) of the components was calculated using the relative area under the peak curve of the HPLC chromatogram.

**Table 2 molecules-27-01582-t002:** Combination index (CI) and dose-reduction index (DRI) of SAA and OSR on H8 cells.

SAA/OSR	4:1	2:1	1:1	1:2	1:4
CI	0.9	0.91	0.8	0.59	0.71
DRI_SAA_	1.22	1.32	1.76	3.1	3.73
DRI_OSR_	11.81	6.4	4.27	3.75	2.26

**Table 3 molecules-27-01582-t003:** IC_50_ values of SAA and OSR, either alone or in a two-drug combination.

	IC_50_ of SAA (μM)	IC_50_ of OSR (μM)
SAA	5.74 ± 0.63	
OSR		13.70 ± 1.71
4:1	4.83 ± 0.17	1.20 ± 0.04
2:1	4.50 ± 0.58	2.25 ± 0.29
1:1	3.26 ± 0.53	3.26 ± 0.53
1:2	1.87 ± 0.37	3.73 ± 0.74
1:4	1.51 ± 0.26	6.05 ± 1.02

## Data Availability

All data generted during the current study are included in this article.

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
