# Peer review of "Inhibitory Effect of Salvia miltiorrhiza Extract and Its Active Components on Cervical Intraepithelial Neoplastic Cells"

_molecules, 2022, doi:10.3390/molecules27051582_

Round 1
Reviewer 1 Report
The authors reported an in-depth analysis on the application of Salvia miltiorrhiza extracts to counteract pre-cancerous conditions caused by HPV.
The authors reviewed the manuscript in a precise and thorough manner following the advice of the Reviewers.
The data collected is solid and well discussed.
In my opinion, the manuscript can be accepted for publication.
Author Response
Thank you very much for your comments and affirmation. We appreciate your time and efforts in dealing with our manuscript.
Reviewer 2 Report
Despite the limitations pointed out in the different versions of the manuscript, the authors satisfactorily responded to most of the recommendations. In this context, I recommend the publication of this work.
Author Response

(The authors gave the same response as above.)

Reviewer 3 Report
Dear authors this is 4th time i am reading your paper and now i know every line of your paper because mostly you don't do any changes in the results. There are a lot of flaws in the paper. This paper is very confusing. the readers will be very confused after reading this paper. There are a lot of errors which should be removed.
- In the title the authors mentioned that "Salvia miltiorrhiza extract and its active components" but in the Material and methods sections the authors have written in a lot of areas "drug-combination" how can a bioactive compound can be a "DRUG" the word "drug" is something else bioactive compound is something else. From section 2.5 til 2.9 remove this word "drug combination".
- Figure 3. section B remove the word drug from this graph too.
- section 2.5: line 114: the authors said that "Drug combinations at a series of concentrations were added" what was the concentration of the compound SAA and OSA? in the paper only ratios are mentioned not concentration.
- The results need to be explained moe so that a reader can understand what is written over there.
- The conclusion should be more attractive.
- This article not only needs a native English writer but also needs a scientific writer who can explain the article in a scientific way.
- Throughout the paper write the word IC50 in a lower superscript.
Round 2
Reviewer 3 Report
The authors addressed all the questions very clearly. I recommend for its publications.
This manuscript is a resubmission of an earlier submission. The following is a list of the peer review reports and author responses from that submission.
Round 1
Reviewer 1 Report
HPV-generated cancer studies should be treated more seriously. In my opinion, it is not enough to demonstrate the action of a phyto-extract on a cellular model to establish its validity (even if only potential) as an oncological therapy.
For the umpteenth time, I ask the authors to include reference drugs for the treatment of HPV-generated tumors in their experiments.
As an example see:
https://www.cancer.gov/about-cancer/treatment/drugs/cervical
Reviewer 2 Report
This manuscript has been constantly improved in each round of review. In my opinion, the study of the individual mechanism of action of the specialized metabolites is essential for a comparison with the synergistic mode of action proposed. It would be valuable to verify the effect of each compound on programmed apoptosis and G2/M-phase arrest. Despite this important limitation in the experimental design and the recommendations made in other versions that have already been satisfactorily reviewed by the authors, I have only one suggestion.1. The resolution of figure 2 is of low quality. The x axis is not identified in English and the numbering is very small.
Reviewer 3 Report
The paper is improved a lot as compared to before, But still, there are some sections that needed to revise.
1. The introduction section didn't provide an explanatory background. OS kindly elaborate it.
2. FIg.1 The X-axis and Y-Axis fonts are different from one another. Both should be written in the same font.
3. There are still a lot of language errors. KIndly check it through some native speaker.